# LANGUAGE MODEL PRE-TRAINING ON TRUE NEGATIVES

## ABSTRACT

Discriminative pre-trained language models (PrLMs) learn to predict original texts from intentionally corrupted ones. Taking the former text as positive and the latter as negative samples, the discriminative PrLM can be trained effectively for contextualized representation. However, though the training of such a type of PrLMs highly relies on the quality of the automatically constructed samples, existing PrLMs simply treat all corrupted texts as equal negative without any examination, which actually lets the resulting model inevitably suffer from the false negative issue where training is carried out on wrong data and leads to less efficiency and less robustness in the resulting PrLMs. Thus in this work, on the basis of defining the false negative issue in discriminative PrLMs that has been ignored for a long time, we design enhanced pre-training methods to counteract false negative predictions and encourage pre-training language models on true negatives, by correcting the harmful gradient updates subject to false negative predictions. Experimental results on GLUE and SQuAD benchmarks show that our counter-false-negative pre-training methods indeed bring about better performance together with stronger robustness.

## 1 INTRODUCTION

Large-scale pre-trained language (PrLM) models are playing an important role in a wide variety of NLP tasks with their impressive empirical performance (Radford et al., 2018; Peters et al., 2018; Devlin et al., 2019; Yang et al., 2019; Lan et al., 2019; Clark et al., 2019). So far, there comes two categories of PrLMs, the generative like GPT (Radford et al., 2018) and BART (Lewis et al., 2020b), which employ a decoder for learning to predict a full sequence, and the discriminative like BERT style of PrLMs which learn to predict the original text from the intentionally corrupted ones. In this work, we focus on the latter category of PrLMs, typically with denoising objectives (also known as masked language modeling, MLM) (Liu et al., 2019; Joshi et al., 2020; Sun et al., 2019). In a denoising objective, a certain percentage of tokens in the input sentence are masked out, and the model should predict those corrupted tokens during the pre-training (Peters et al., 2018; Sun et al., 2019; Levine et al., 2021; Li & Zhao, 2021).

Besides corrupting the texts with masks, some alternatives were proposed for constructing training examples with various arbitrary noising functions motivated by edit operations like insertion, deletion, replacement, permutation, and retrieval (Lewis et al., 2020a; Xu & Zhao, 2021; Wang et al., 2019; Guu et al., 2020). Auxiliary objectives are also proposed in conjunction with MLM, such as next sentence prediction (Devlin et al., 2019), span-boundary objective (Joshi et al., 2020), and sentence-order prediction (Lan et al., 2019).

Although existing studies have made progress in designing effective masking strategies and auxiliary objectives, there are intrinsic yet critical issues appearing throughout the whole training process that lack attention for a long time. Discriminative PrLM can be regarded as a kind of auto denoising encoder on automatically corrupted texts. Thus, it is critical to ensure the auto-constructed data is true enough. Intuitively, a discriminative PrLM learns to distinguish two types of samples, positive (already existing original ones) and negative (the corrupted ones from the auto constructing). Taking MLM as an example, a proportion of tokens in sentences are corrupted, e.g., replaced with mask symbols, which would affect the sentence structures, leading to the loss of semantics and increasing the uncertainty of predictions. In extreme cases, such corrupted text may be linguistically correct.

| Example | Ground-truth | Prediction | MLM | Mediation |
|---|---|---|---|---|
| It is [MASK] good | very | happy | - | - |
| The cat is [MASK] | cute | smart | ✗ | ✓ |
| It is a [MASK] [MASK] for discussion | good day | great time | ✗ | ✓ |

Table 1: Examples of true negative (the first line) and false negatives (the second and third line).

However, the current PrLMs simply consider all corrupted texts as negative samples, so that the resulting PrLM has to be trained on such false negatives with less efficiency and less robustness, which either waste training time on meaningless data or are vulnerable to adversarial attacks like diversity distraction and synonym substitution (Wang et al., 2021).

In a general scenario, MLM only calculates label-wise matching between the prediction and the gold tokens in the training process, thus inevitably suffering from the issue of false negatives where the prediction is meaningful but regarded as wrong cases, as examples shown in Table 1. The issue is also observed in sequence generation tasks, which is tied to the standard training criterion of maximum likelihood estimation (MLE) that treats all incorrect predictions as being equally incorrect (Wieting et al., 2019; Li et al., 2020). Instead of measuring negative diversity via the diversity scores between the different incorrect model outputs, our method is dedicated to mediating the training process by detecting the alternative predictions as opposed to the gold one, to steer model training on true negatives, which benefits the resulting language modeling in general.

Though the false negatives may potentially hurt the pre-training in both efficiency and robustness to a great extent, it is surprising that this problem is kept out of the research scope of PrLMs until this work to our best knowledge. To address the issue of misconceived false negative predictions and encourage pre-training language models on true negatives or more true negatives, we present an enhanced pre-training approach to counteract misconceived negatives. In detail, we employ two enhanced pre-training objectives: 1) soft regularization by minimizing the semantic distances between the prediction and the original one to smooth the rough cross-entropy and 2) hard correction to shield the gradient propagation of the false negative samples to avoid training with false negative predictions. We pre-train our methods on top of the ELECTRA architecture (Clark et al., 2019) and fine-tune it on widely-used down-streaming benchmark tasks, including GLUE (Wang et al., 2018) and SQuAD (Rajpurkar et al., 2016). Experimental results show that our approach boosts the baseline performance by a large margin, which verifies the effectiveness of our proposed methods and the importance of training on true negatives. Case studies show that our method keeps the simplicity and also improves the robustness of language model pre-training.

## 2 RELATED WORK

Designing effective criteria for language modeling is one of the major topics in training pre-trained models, which decides how the model captures knowledge from large-scale unlabeled data. Recent studies have investigated denoising patterns (Raffel et al., 2020; Lewis et al., 2020b), MLM alternatives (Yang et al., 2019), and auxiliary objectives (Lan et al., 2019; Joshi et al., 2020) to improve the power of pre-training. However, studies show that the current models still suffer from under-fitting issues, and it remains challenging to find effective and efficient training strategies (Rogers et al., 2020).

**Denoising Patterns** MLM has been widely used as the major objective for pre-training (Devlin et al., 2019; Lan et al., 2019; Clark et al., 2019; Song et al., 2020), in which the fundamental part is how to construct high-quality masked examples (Raffel et al., 2020). The current studies commonly define specific patterns for mask corruption. For example, some are motivated from the language modeling units, such as subword masking (Devlin et al., 2019), span masking (Joshi et al., 2020), and $n$-gram masking (Levine et al., 2021; Li & Zhao, 2021). Some employ a variety of edit operations like insertion, deletion, replacement, and retrieval (Lewis et al., 2020a; Guu et al., 2020). Others seek for external knowledge annotations, such as named entities (Sun et al., 2019), semantics (Zhou et al., 2020), and syntax (Zhang et al., 2020b; Xu et al., 2021). To provide more diversity of mask tokens, RoBERTa applied dynamic masks in different training iterations (Liu et al., 2019). These

prior studies either employ pre-defined mask construction patterns or improve the diversity of mask tokens to help capture the knowledge from pre-training.

**MLM alternatives** To alleviate the task mismatch between the pre-training and the fine-tuning for downstream tasks, XLNet (Yang et al., 2019) proposed an autoregressive objective for language modeling through token permutation, which further adopts a more complex model architecture. Instead of corrupting sentences with the mask symbol that never appears in the fine-tuning stage, MacBERT (Cui et al., 2020) propose to use similar words for the masking purpose. Yamaguchi et al. (2021) also investigates simple pre-training objectives based on token-level classification tasks as replacements of MLM, which are often computationally cheaper and result in comparable performance to MLM. In addition, training sequence-to-sequence (Seq2Seq) language models has also aroused continuous interests (Dong et al., 2019; Lewis et al., 2020b; Raffel et al., 2020).

**Auxiliary objectives** Another research line is auxiliary objectives in conjunction with MLM, such as next sentence prediction (Devlin et al., 2019), span-boundary objective (Joshi et al., 2020), and sentence-order prediction (Lan et al., 2019). Such line of researches emerges as hot topics, especially in domain-specific pre-training, such as dialogue-oriented language models, which involve diverse kinds of interaction entailed in utterances (Zhang et al., 2020a; Wu et al., 2020; Zhang & Zhao, 2021).

As the major difference from the existing studies, our work devotes itself to mediating misconceived negatives as the essential drawback of MLM during the MLE estimation and aiming to guide language models to learn from true negatives through our newly proposed regularization and correction methods. The comparison with existing work is illustrated in Figure 1.

Besides the heuristic pre-trained patterns like masking strategies during data construction, we stress that there are potential post-processing strategies to guide the MLM training: correction and pruning, which are considered to deal with the false negative issue during MLM training, where the model would yield reasonable predictions but discriminated as wrong predictions because such predictions do not match the single gold token for each training case. For example, many tokens are reasonable but written in different forms or are the synonyms of the expected gold token. We could correct the training with soft regularization or directly drop

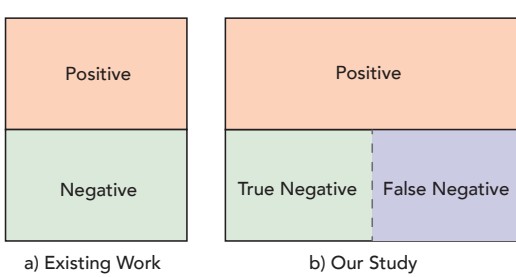

a) Existing Work          b) Our Study

Figure 1: Overview of our study.

the uncertain predictions. Promoting our view to sentence level, the similarity between the predicted sentence and the original sentence can also be taken into account to measure the sentence-level confidence that indicates how hard the task is, which would be beneficial to provide more fine-grained signals and thus improve the training quality. Based on the rationales above, we are motivated to design the corresponding correction and regularization techniques to mediate misconceived negatives.

In a broader view, our work is also related to knowledge distillation, whose paradigm is training student networks to mimic the soft target generated by well-trained teachers (Gou et al., 2021; Hahn & Choi, 2019), which has been proved as a type of label smoothing (Yuan et al., 2020; Zhang & Sabuncu, 2020; Müller et al., 2019). Such a line of research has supported the hypothesis that regularization of soft targets could accelerate convergence and promote performance. By contrast, our approaches are more efficient without the need to train two models.

## 3 METHODOLOGY

### 3.1 PRELIMINARIES

**MLM** Masked LM (MLM) is a denoising language model technique used by BERT (Devlin et al., 2019) to take advantage of both the left and right contexts. Given a sentence $\mathbf{s} = \{w_1, w_2, \ldots, w_n\}$, A certain proportion of tokens are randomly replaced with a special mask symbol. The input is

fed into the multi-head attention layer to obtain the contextual representations, which is defined as $H = \text{FFN}(\text{MultiHead}(K, Q, V))$, where K,Q,V are packed from the input sequence representation $\mathbf{s}$. Then, the model is trained to predict the masked token based on the context.

Let $\mathcal{Y} \in R^{N_m}$ denote the set of masked positions using the mask symbol $[M]$ and $N_m$ is the number of masked tokens. We have $w_k \in \mathcal{Y}$ as the set of masked tokens, and $\mathbf{s}'$ as the masked sentence where the tokens in $\mathcal{Y}$ are masked with the mask symbol in $s$. The objective of MLM is to maximize the following objective:

$$\mathcal{L}_{mlm}(w_k, \mathbf{s}') = -\frac{1}{N_m} \sum_{k \in \mathcal{Y}}^{N_m} \log p_\theta(w_k \mid \mathbf{s}'), \tag{1}$$

where $N_m$ is the total number of masked positions in the input sequence.

**ELECTRA**  MLM only learns from a small proportion of masked positions per example, which incur a substantial compute cost. With the goal to improve training efficiency, ELECTRA (Clark et al., 2019) is proposed, which consists of a generator $G$ and a discriminator $D$. Instead of masking tokens, ELECTRA corrupts the input sequence by replacing them with tokens sampled from a small generator. The discriminator is trained to distinguish whether each replaced one is the original or a replacement. In the implementation, the generator and discriminator are based on Transformer architecture like BERT (Devlin et al., 2019) but mainly differ in the model size. The generator is in a smaller scale, and the training objective is the same as MLM in Eq. 1, written as:

$$\mathcal{L}_G(w_k, \mathbf{s}') = -\frac{1}{N_m} \sum_{k \in \mathcal{Y}}^{N_m} \log p_\theta^G(w_k \mid \mathbf{s}'). \tag{2}$$

During the training iteration, the generator predicts a new sequence (denoted as $\mathbf{s}^g$) with the predicted token for each corrupted position in the original sequence. The predicted sequence is then fed to the discriminator, which uses a binary classification task to predict the probability $D(w_t^r, \mathbf{s}^g)$ to indicate how likely each token $w_t^r (t \in [1, n])$ in $\mathbf{s}^g$ is replaced by generator, whose loss function is:

$$\begin{aligned}
\mathcal{L}_D(w_t, \mathbf{s}') &= \frac{1}{n} \sum_{t=1}^{N} \mathcal{L}(w_t^r, \mathbf{s}^g), \\
\mathcal{L}(w_t^r, \mathbf{s}^g) &= \begin{cases} -\log D(w_t^r, \mathbf{s}^g), & w_t^r = w_t \\ -\log(1 - D(w_t^r, \mathbf{s}^g)), & w_t^r \neq w_t \end{cases}
\end{aligned} \tag{3}$$

where $n$ is the length of the input sequence.

The final combined loss of ELECTRA is computed by: $\mathcal{L}_{dlm} = \mathcal{L}_G(w_k, \mathbf{s}') + \lambda \mathcal{L}_D(w_t, \mathbf{s}')$, where $\lambda$ is a hyper-parameter to balance the weights of generator and discriminator, and it is set to 50 according to Clark et al. (2019).

## 3.2 PRE-TRAINING ON TRUE NEGATIVES

An intuitive solution to encourage the language model pre-training on true negatives is to reduce the "difficulty" or uncertainty of the prediction during pre-training. Therefore, the cloze-style token prediction problem may be simplified as a multi-choice problem to break the rough validation between the prediction and ground-truth and smooth measurement of the relevance. Therefore, we are motivated to employ two techniques to counteract the false negative predictions, including 1) soft regularization, which measures the distribution similarity between the predicted token and the original one, to smooth the tough cross-entropy by minimizing the semantic distances (SR); 2) hard correction (HC), which shields the gradient propagation of the false negative samples to further avoid training with false negative predictions.

**Soft Regularization**  Let $p_k$ denote the predicted token from MLM (derived from the generator in this work). For $w_k$ and $p_k$, we fetch their token representations from the model's embedding module,[1] denoted as $e_k$ and $e_k'$, respectively. We leverage cosine similarity as the regularization

---

[1]For ELECTRA, we fetch the embedding from the discriminator. Note that the embeddings of the generator and the discriminator are tied following the official implementation (Clark et al., 2019).

based on the intuition that the semantic distance between the prediction and gold tokens should be minimized:

$$\mathcal{L}_{reg} = \sum_{k=1}^{m}(1 - \frac{e_k \cdot e_k'}{\|e_k\| \cdot \|e_k'\|}). \tag{4}$$

SR is based on the hypothesis that the predicted tokens should have a semantic relationship with the gold ones in the embedding space to some extent, which is supported by various existing studies (Bordes et al., 2013; Zhang & Zhao, 2021; Chen et al., 2021; Li et al., 2020). We choose to apply SR to the embedding layer because the embedding layer is the most fundamental and stable layer. Optimizing the embedding layer would possibly lead to a more severe influence of the model training and help the model learn semantics between words better as indicated by Jiang et al. (2020).

**Hard Correction**   The other alternative strategy is to prune the gradient when the model suffers from the confusion of whether the prediction is correct or not. For each prediction, we check if the token is highly related to the ground-truth token based on a short lookup table $\mathcal{V}$ in which each token is mapped to a list of alternatives. The lookup table is built by retrieving the synonym alternatives for each word in the model vocabulary, e.g., from WordNet (Miller, 1995) or Word2Vec embedding (Mikolov et al., 2013). In this work, we use WordNet synonyms by default (Section 5.2 will compare retrieving synonyms from WordNet and Word2Vec embedding).

$$\mathcal{L}_{cor} = \sum_{k \in \mathcal{V}}^{m} \boldsymbol{I}_k * \log p_{\theta}(w_k \mid \mathbf{s}'), \tag{5}$$

where $\boldsymbol{I}_k$ is the identifier indicating whether the $k$-th prediction should be counted, which is defined by:

$$\boldsymbol{I}_k = \begin{cases} 0 & e_k' \neq e_k, e_k \in \mathcal{V}[e_k'], \\ 1 & \text{otherwise.} \end{cases} \tag{6}$$

For each training iteration, if the gold token is found in the synonym list for the predicted token, then the correction is activated by $\boldsymbol{I}_k$. Such a prediction will be judged as correct by HC in cross-entropy — the correction can be applied by simply ignoring this prediction before feeding to the cross-entropy loss function.

### 3.3   IMPLEMENTATION VARIANT

According to the motivation and formulation above, the soft regularization and hard correction approaches are supposed to be applied as independent substitutes.[2] Therefore, the overall training objective for language modeling is rewritten as $\mathcal{L}' = \mathcal{L}_{dlm} + \mathcal{L}_{reg}$ or $\mathcal{L}' = \mathcal{L}_{dlm} + \mathcal{L}_{cor}$ for SR and HC, respectively.

## 4   EXPERIMENTS

### 4.1   SETUP

**Pre-training**   In this part, we will introduce the model architecture, hyper-parameter setting, and corpus for pre-training our models. Considering the training efficiency, we employ ELECTRA small and base as our backbone models and implement our pre-training objectives on top of them. We follow the model configurations in Clark et al. (2019) for fair comparisons. For hyper-parameters, the batch size is 128 for the base models in our work instead of 256 as in the original setting due to limited resources. The mask ratio is 15%. We set a maximum number of tokens as 128 for small models and 512 for base models.[3] The small models are pre-trained from scratch for 1000$k$ steps. To save computation, like previous studies (Dong et al., 2019), we continue training base models for 200$k$ steps using the pre-trained weights as initialization. The learning rates for small and base

---

[2]We find that combining the two strategies would not yield a clear advantage over each individual. The possible reason would be the redundancy which may lead to similar effects.

[3]For evaluation on the reading comprehension tasks, we also pre-train the variants with the length of sentences in each batch as up to 512 tokens.

| Model | CoLA | SST | MRPC | STS | QQP | MNLI | QNLI | RTE | Average | Δ |
|---|---|---|---|---|---|---|---|---|---|---|
| *Single model on dev set* | | | | | | | | | | |
| $\text{ELECTRA}_{\text{small}}$ | 56.8 | 88.3 | 87.4 | 86.8 | 88.3 | 78.9 | 87.9 | 68.5 | 80.4 | - |
| $\text{ELECTRA}^{\text{SR}}_{\text{small}}$ | 61.1 | **90.1** | **89.5** | **87.0** | **89.4** | **80.8** | **88.8** | **68.6** | **81.9** | ↑1.5 |
| $\text{ELECTRA}^{\text{HC}}_{\text{small}}$ | **62.0** | 89.8 | 87.0 | 86.7 | 89.0 | 80.4 | 88.0 | 67.9 | 81.4 | ↑1.0 |
| $\text{ELECTRA}_{\text{base}}$ | 68.3 | 95.3 | 90.9 | **91.3** | 91.7 | 88.5 | 93.0 | 82.3 | 87.7 | - |
| $\text{ELECTRA}^{\text{SR}}_{\text{base}}$ | 70.4 | 95.4 | 90.4 | 91.2 | 91.9 | **89.1** | 93.4 | **84.8** | 88.3 | ↑0.6 |
| $\text{ELECTRA}^{\text{HC}}_{\text{base}}$ | **70.9** | **95.6** | **91.2** | 91.3 | **92.0** | 88.7 | **93.6** | 83.8 | **88.4** | ↑0.7 |
| *Single model on test set* | | | | | | | | | | |
| $\text{ELECTRA}_{\text{small}}$ | 52.3 | 89.7 | 84.8 | 80.5 | **88.4** | 79.9 | 88.0 | 62.9 | 78.3 | - |
| $\text{ELECTRA}^{\text{SR}}_{\text{small}}$ | **58.3** | **90.6** | **85.4** | 81.4 | 87.9 | **80.6** | 88.0 | **64.3** | **79.6** | ↑1.3 |
| $\text{ELECTRA}^{\text{HC}}_{\text{small}}$ | 55.3 | 90.3 | 84.1 | **82.0** | 87.2 | **80.6** | **88.4** | **64.3** | 79.0 | ↑0.7 |
| $\text{ELECTRA}_{\text{base}}$ | 62.4 | 95.3 | 87.3 | 89.9 | 89.6 | 88.6 | 93.4 | 78.1 | 85.6 | - |
| $\text{ELECTRA}^{\text{SR}}_{\text{base}}$ | 65.7 | 95.7 | 88.3 | **90.0** | **89.9** | **89.1** | **93.6** | 78.8 | 86.4 | ↑0.8 |
| $\text{ELECTRA}^{\text{HC}}_{\text{base}}$ | **67.5** | **95.8** | **88.6** | 89.9 | 89.7 | 89.0 | **93.6** | **79.1** | **86.7** | ↑1.1 |

Table 2: Comparisons between our proposed methods and the previous strong pre-trained models under small and base setting on the dev and test set of GLUE tasks. STS is reported by Spearman correlation, CoLA is reported by Matthew's correlation, and other tasks are reported by accuracy.

models are 5e-4, and 5e-5, respectively. We use OpenWebText (Radford et al., 2019) to train small models, and Wikipedia and BooksCorpus (Zhu et al., 2015) for training base models following Clark et al. (2019).[4] The baselines are trained to the same steps for a fair comparison.

**Fine-tuning** For evaluation, we fine-tune the pre-trained models on GLUE (General Language Understanding Evaluation) (Wang et al., 2018) and SQuAD v1.1 (Rajpurkar et al., 2016) to evaluate the performance of the pre-trained models. GLUE include two single-sentence tasks (CoLA (Warstadt et al., 2018), SST-2 (Socher et al., 2013)), three similarity and paraphrase tasks (MRPC (Dolan & Brockett, 2005), STS-B (Cer et al., 2017), QQP (Chen et al., 2018) ), three inference tasks (MNLI (Nangia et al., 2017), QNLI (Rajpurkar et al., 2016), RTE (Bentivogli et al., 2009). We follow ELECTRA hyper-parameters for single-task fine-tuning. We did not use any training strategies like starting from MNLI, to avoid extra distractors and focus on the fair comparison in the single-model and single-task settings.

## 4.2 RESULTS

We evaluate the performance of our pre-training enhancement compared with the baselines in small and base sizes on GLUE and SQuAD benchmarks in Tables 2-3. From the results, we have the following observations:

1) The models with our enhanced pre-training objectives outperform the baselines in all the subtasks. With the same configuration and pre-training data, for both the small-size and the base-size, our methods outperform the strong ELECTRA baselines by +1.5(dev)/+1.3(test) and +0.7(dev)/+1.1(test) on average, respectively. The results demonstrate that our pro-

| Model | Exact Match | F1 Score |
|---|---|---|
| $\text{ELECTRA}_{\text{small}}$ | 75.8 | 83.9 |
| $\text{ELECTRA}^{\text{SR}}_{\text{small}}$ | 76.0 (↑0.2) | 84.2 (↑0.3) |
| $\text{ELECTRA}^{\text{HC}}_{\text{small}}$ | **77.7** (↑1.9) | **85.6** (↑1.7) |
| $\text{ELECTRA}_{\text{base}}$ | 85.1 | 91.6 |
| $\text{ELECTRA}^{\text{SR}}_{\text{base}}$ | 85.6 (↑0.5) | 92.0 (↑0.4) |
| $\text{ELECTRA}^{\text{HC}}_{\text{base}}$ | **85.7** (↑0.6) | **92.1** (↑0.5) |

Table 3: Results on the SQuAD dev set.

posed methods improve the pre-training of ELECTRA substantially and disclose that mediating the training with true negatives is quite beneficial for improving language model pre-training. To verify

---

[4]Our codes and models will be publicly available.

| Model | Params | CoLA | SST | MRPC | STS | QQP | MNLI | QNLI | RTE | Avg. | $\Delta$ |
|---|---|---|---|---|---|---|---|---|---|---|---|
| $BERT_{base}$ | 110M | 52.1 | 93.5 | 84.8 | 85.8 | 89.2 | 84.6 | 90.5 | 66.4 | 80.9 | - |
| $BERT_{large}$ | 335M | 60.5 | 94.9 | 85.4 | 86.5 | 89.3 | 86.7 | 92.7 | 70.1 | 83.3 | - |
| $SpanBERT_{large}$ | 335M | 64.3 | 94.8 | 87.9 | 89.9 | 89.5 | 87.7 | 94.3 | 79.0 | 85.9 | - |
| $ELECTRA_{small}$ | 14M | 54.6 | 89.1 | 83.7 | 80.3 | 88.0 | 79.7 | 87.7 | 60.8 | 78.0 | - |
| $ELECTRA_{base}$ | 110M | 59.7 | 93.4 | 86.7 | 87.7 | 89.1 | 85.8 | 92.7 | 73.1 | 83.5 | - |
| $ELECTRA_{small}^{SR}$ | 14M | 58.3 | 90.6 | 85.4 | 81.4 | 87.9 | 80.6 | 88.0 | 64.3 | 79.6 | ↑1.6 |
| $ELECTRA_{small}^{HC}$ | 14M | 55.3 | 90.3 | 84.1 | 82.0 | 87.2 | 80.6 | 88.4 | 64.3 | 79.0 | ↑1.0 |
| $ELECTRA_{base}^{SR}$ | 110M | 65.7 | 95.7 | 88.3 | **90.0** | **89.9** | **89.1** | 93.6 | 78.8 | 86.4 | ↑2.9 |
| $ELECTRA_{base}^{HC}$ | 110M | **67.5** | **95.8** | **88.6** | 89.9 | 89.7 | 89.0 | **93.6** | **79.1** | **86.7** | ↑3.2 |

Table 4: Comparisons with public methods on GLUE test sets. The public results are from BERT (Devlin et al., 2019), SpanBERT (Joshi et al., 2020), and ELECTRA (Clark et al., 2019).

| Model | CoLA | SST | MRPC | STS | QQP | MNLI | QNLI | RTE | Average | $\Delta$ |
|---|---|---|---|---|---|---|---|---|---|---|
| $ELECTRA_{small}$ | 56.8 | 88.3 | 87.4 | 86.8 | 88.3 | 78.9 | 87.9 | 68.5 | 80.4 | - |
| $ELECTRA_{Word}^{SR}$ | 61.1 | 90.1 | 89.5 | 87.0 | 89.4 | 80.8 | 88.8 | 68.6 | 81.9 | ↑1.5 |
| $ELECTRA_{Sent}^{SR}$ | 59.5 | 89.6 | 90.0 | 86.7 | 89.1 | 80.4 | 90.0 | 68.2 | 81.6 | ↑1.2 |
| $ELECTRA_{WordNet}^{HC}$ | 62.0 | 89.8 | 87.0 | 86.7 | 89.0 | 80.4 | 88.0 | 67.9 | 81.4 | ↑1.0 |
| $ELECTRA_{Embedding}^{HC}$ | 59.0 | 88.5 | 87.0 | 86.4 | 88.8 | 79.6 | 87.9 | 67.1 | 80.6 | ↑0.2 |

Table 5: Comparative studies of variants on GLUE dev sets based on small models. The first block compare the word-level regularization and sentence-level regularization, respectively. The second block shows the results of HC methods based on WordNet and Word2Vec embedding, respectively.

the generality of our methods, we also implement them on BERT backbones as details presented in Appendix A.2. The results show that our methods achieve consistent gains.

2) Table 4 shows the comparison with public models on the GLUE test set. Compared with the public methods, our model not only far exceeds the performance of others under the same model scale, but also outperforms the larger models with much fewer parameters.

3) The performance gains on the small-size models are more obvious than the base-size models. We speculate that is due to the learning of the small-size generator is more insufficient and suffers from the false negative issue more seriously.

4) Both SR and HC pre-training strategies help the resulting model surpass the baselines obviously. Note that our proposed method is model-agnostic so that the convenient usability of its backbone precursor can be kept without architecture modifications. In comparison, SR is more generalizable as it does not require extra resources, while HC has the advantage of interpretation via explicit correction.

5) Our enhanced pre-training objectives show considerable performance improvements on linguistics-related tasks such as CoLA and MRPC. These tasks are about linguistic acceptability and paraphrase/semantic equivalence relationship. In addition, our methods also achieve obvious gains in tasks requiring more complex semantic understanding and reasoning, such as MNLI and SQuAD, showing that they may help capture semantics to some extent.

6) Our methods are lightweight that keep nearly the same parameter size, computation requirement, and training speed as the baseline but with stronger capacity.

# 5 ANALYSIS

## 5.1 WORD-LEVEL REGULARIZATION VS. SENTENCE-LEVEL REGULARIZATION

The soft regularization approach measures the semantic distance between the predicted one and the ground-truth, which may neglect the sentence-level context (though the token representation may

have already captured contextualized representation to some extent). We are interested in whether measuring the sentence-level similarity would achieve even better results. To verify the hypothesis, we fill the masked sentence $\mathbf{s}'$ with the predicted tokens $e'_k$ to have the predicted sentence $s_p$. Then, $s_p$ and $s$ are fed to the Transformer encoder to have the contextualized representation $H_p$ and $H_s$, respectively. To guide the probability distribution of model predictions $H_p$ to match the expected probability distribution $H_s$, we adopt Kullback–Leibler (KL) divergence:

$$\mathcal{L}_{kl} = \mathrm{KL}(H_p \parallel H_s), \tag{7}$$

where $\mathcal{L}_{kl}$ is applied as the degree to reflect the sentence level semantic mismatch. The loss function is then written as $\mathcal{L}' = \mathcal{L}_{dlm} + \mathcal{L}_{kl}$.

For clarity, we denote the original $\mathrm{ELECTRA}^{\mathrm{SR}}_{\mathrm{small}}$ method described in Eq. 4 as $\mathrm{ELECTRA}^{\mathrm{SR}}_{\mathrm{Word}}$ and the sentence-level variant as $\mathrm{ELECTRA}^{\mathrm{SR}}_{\mathrm{Sent}}$. The comparative results are reported in the first block of Table 5, which indicates that using sentence-level regularization ($\mathrm{ELECTRA}^{\mathrm{SR}}_{\mathrm{Sent}}$) also substantially outperforms the baseline and nearly reaches the performance of word-level one ($\mathrm{ELECTRA}^{\mathrm{SR}}_{\mathrm{Word}}$) on average, with slightly better results on MRPC and MNLI. Although $\mathrm{ELECTRA}^{\mathrm{SR}}_{\mathrm{Sent}}$ still keeps the same parameter size with baseline, it leads to more computation resources because it requires the extra calculation of the contextualized representation for the predicted token sequence $H_p$. Therefore, considering the balance between effectiveness and efficiency, $\mathrm{ELECTRA}^{\mathrm{SR}}_{\mathrm{Word}}$ can serve as the first preferred choice for practical applications, and $\mathrm{ELECTRA}^{\mathrm{SR}}_{\mathrm{Sent}}$ can be employed when computation resources are sufficient.

## 5.2 RETRIEVING SYNONYMS FROM WORDNET VS. WORD2VEC EMBEDDINGS

For the hard correction approach, the candidate synonyms for detecting false negative predictions can be derived from WordNet (Miller, 1995) or Word2Vec embedding space (Mikolov et al., 2013) as described in Section 3.2.[5] To verify the impact of different sources, we compare the results as shown in the second block of Table 5. We see that $\mathrm{ELECTRA}^{\mathrm{HC}}_{\mathrm{WordNet}}$ outperforms $\mathrm{ELECTRA}^{\mathrm{HC}}_{\mathrm{Embedding}}$ by a large margin. The most plausible reason would be that the retrieved list of synonyms from $\mathrm{ELECTRA}^{\mathrm{HC}}_{\mathrm{WordNet}}$ would have higher quality than that from $\mathrm{ELECTRA}^{\mathrm{HC}}_{\mathrm{Embedding}}$. Although the embedding-based method may benefit from semantic matching, but would also bring noises as it is hard to set the threshold to ensure the top-ranked words are accurate synonyms. Therefore, $\mathrm{ELECTRA}^{\mathrm{HC}}_{\mathrm{WordNet}}$ turns out to be better suitable for our task.

To interpret how our method works, we randomly select some semantic correction examples as shown in Figure 2 by taking the baseline as the backbone model. We find that the baseline model produces reasonable predictions such as *main*, *remain*, *attempt* as opposed to the golds ones, *primary*, *stay*, *effort*. Those predictions will be determined as wrong and then harm pre-training. Fortunately, such cases can be easily solved by our proposed method.

## 5.3 ROBUSTNESS EVALUATION

Intuitively, our method would be helpful for improving the robustness of the pre-trained models because the approaches may indicate lexical semantics and representation diversity during the correction or regularization operations. To verify the hypothesis, we use a robustness evaluation platform TextFlint (Wang et al., 2021) on SQuAD, from which two standard transformation methods are adapted: 1) *AddSentenceDiverse* generates distractors with altered questions and fake answer sand 2) *SwapSynWordNet* transforms an input by replacing its words with synonyms provided by WordNet.

Table 6 shows the robustness evaluation results. We observe that both kinds of attacks induce a significant performance drop of the baseline system, by 54.95% and 6.0% on the EM metrics, respectively, indicating that the system is sensitive to distractors with similar meanings. In contrast, both of our models can effectively resist those attacks with less performance degradation. Specifically, the HC method works stably in the *SwapSynWordNet* attack. We speculate the reason is that the hard correction strategy models the synonym information during pre-training, which would help

---

[5]Since the embedding method returns a ranked list by calculating the similarity score with the whole vocabulary, we only take the top 10 most similar words for each retrieval.

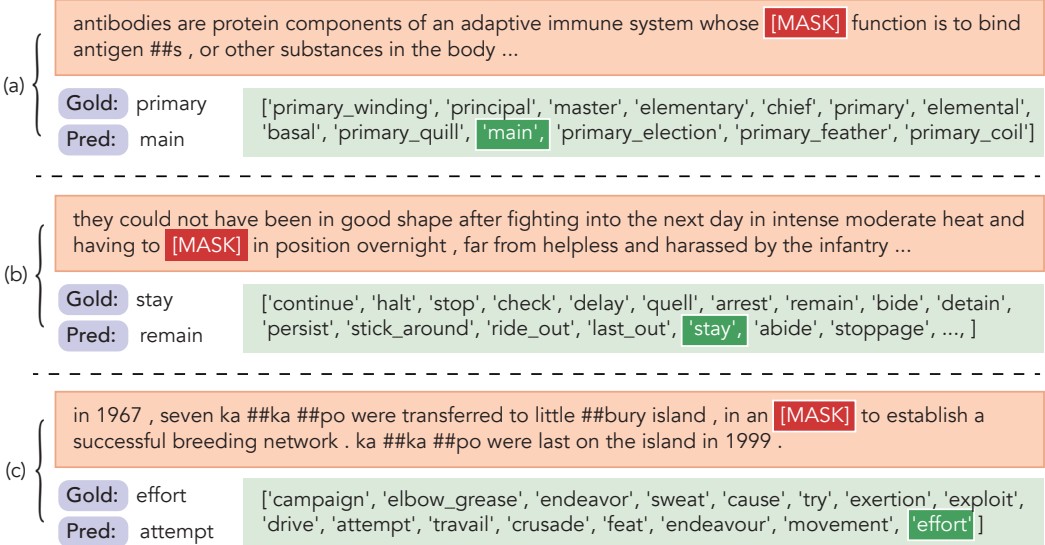

Figure 2: Interpretation of the semantic correction process. The orange box contain the input sentence, the blue buttoms indicate the gold and predicted tokens, and the green box shows the candidate synonyms from WordNet given the predicted token.

| Model | AddSentenceDiverse (Ori.→Trans.) | | SwapSynWordNet (Ori.→Trans.) | |
| | Exact Match | F1 Score | Exact Match | F1 Score |
| --- | --- | --- | --- | --- |
| ELECTRA$_{\text{small}}$ | 80.55→25.60 (↓54.95) | 85.10→26.43 (↓58.67) | 80.67→74.67 (↓6.00) | 85.38→80.43 (↓4.95) |
| ELECTRA$_{\text{small}}^{\text{SR}}$ | 78.84→**37.20** (↓**41.64**) | 80.84→**38.29** (↓**42.55**) | 78.67→75.67 (↓3.00) | 80.88→78.51 (↓**2.37**) |
| ELECTRA$_{\text{small}}^{\text{HC}}$ | 82.59→34.13 (↓48.46) | 86.78→36.60 (↓50.18) | 82.33→**79.67** (↓**2.66**) | 86.68→**83.65** (↓3.03) |

Table 6: Robustness evaluation on the SQuAD dataset. Ori. represents the results of original dataset derived from the SQuAD 1.1 dev set by TextFlint (Wang et al., 2021) while Trans. indicates the transformed one. The assessed models are the small models from Table 3.

capture lexical semantics. The other variant, the soft regularization objective, achieves much better performance in the *AddSentenceDiverse*. The most plausible reason might be the advantage of acquiring semantic diversity by regularizing the semantic distance in the SR objective. The results indicate that both methods achieve similar effects of robustness in general but also have some slight emphasis.

## 6 CONCLUSIONS

Though discriminative PrLMs may quite straightforwardly suffer from the false negative issue according to our exploration in this work, it has been completely ignored for a long time and it is a bit surprising that maybe this work is the first one that formally considers such a big pre-training leak. To counteract the intrinsic and critical issue, we employ extra pre-training objectives to correct or prune the harmful gradient update after detecting the false negative predictions. Experimental results on GLUE and SQuAD benchmarks verify the superiority of our pre-training enhancement. Robustness evaluation shows that our methods can help the resulting PrLM effectively resist various attacks while existing common PrLMs would suffer from significant performance degradation. To our best knowledge, it is also the first work to consider model effectiveness and robustness of language model pre-training at the same time. Our work indicates that mediating false negatives is so important that counter-false-negative pre-training can indeed synchronously improve the effectiveness and robustness of PrLMs.

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

# A APPENDIX

## A.1 WOULD THE HARD CORRECTION BRING FALSE POSITIVES?

The hard correction would not bring false positives because it is a post-processing technique. As described in the last paragraph of Section 3.2, if the predicted token is in the shortlist, the correction will be activated by simply ignoring this prediction before feeding to cross-entropy loss function. In our PyTorch implementtaion, for example, the corresponding gold label id will be replaced by -100 (default ignore_index in nn.CrossEntropy using PyTorch), which means this token is not required to predict anymore.

| Model | CoLA Mcc | SST Acc | MRPC Acc | STS Spear | QQP Acc | MNLI Acc | QNLI Acc | RTE Acc | Average - |
|---|---|---|---|---|---|---|---|---|---|
| BERT$_{base}$ | 61.09 | 93.00 | 86.76 | 87.09 | 90.79 | 84.72 | 91.42 | 67.87 | 82.84 |
| BERT$_{base}^{SR}$ | 61.17 | 93.46 | 88.97 | 87.45 | 90.93 | 84.83 | 91.62 | 68.59 | 83.38 |
| BERT$_{base}^{HC}$ | 62.88 | 93.23 | 87.50 | 87.41 | 90.92 | 84.92 | 91.54 | 69.31 | 83.46 |
| BERT$_{Large}$ | 61.67 | 93.69 | 88.48 | 90.14 | 91.30 | 86.74 | 92.37 | 72.92 | 84.67 |
| BERT$_{large}^{SR}$ | 62.26 | 94.15 | 89.22 | 90.12 | 91.41 | 87.01 | 92.82 | 74.01 | 85.13 |
| BERT$_{large}^{HC}$ | 62.34 | 93.35 | 88.97 | 90.48 | 91.46 | 86.96 | 92.95 | 73.65 | 85.02 |

Table 7: Results of BERT methods under base and large setting on the GLUE dev sets. STS is reported by Spearman correlation, CoLA is reported by Matthew's correlation, and other tasks are reported by accuracy.

## A.2 COULD THIS METHOD BE APPLIED ON OTHER MLM PRLMS?

To verify the generality of our methods on other PrLMs, we implemented them on BERT$_{base}$ and BERT$_{large}$ backbones (Devlin et al., 2019) following the same implementation for ELECTRA$_{base}$ as described in Section 4.1. Specifically, we train MLM with our methods based on BERT$_{base}$ and BERT$_{large}$ checkpoints for $200k$ steps on the Wikipedia and BooksCorpus, and fine-tune them on

| Checkpoint (base) | Iteration | Prediction | Checkpoint (large) | Iteration | Prediction |
|---|---|---|---|---|---|
| 6.25% | 6.90% | 1.31% | 6.25% | 7.46% | 1.5% |
| 12.5% | 6.96% | 1.34% | 12.5% | 7.58% | 1.55% |
| 25.0% | 6.97% | 1.36% | 25.0% | 7.31% | 1.49% |
| 50.0% | 7.05% | 1.36% | 50.0% | 7.46% | 1.56% |
| 80.0% | 7.06% | 1.40% | 80.0% | 7.38% | 1.57% |
| 100.0% | 7.07% | 1.41% | 100.0% | 7.44% | 1.60% |

Table 8: Statistics of the hard corrections under base and large settings on the wikitext-2-raw-v1 corpus. Checkpoint means the checkpoint saved at the specific training steps (%).

GLUE tasks. For fair comparison, we train the baseline models based on the same checkpoint in the same manner. Results in Table 7 show that our methods achieve consistent gains on BERT methods.

### A.3 STATISTICS OF THE HARD CORRECTIONS

To have an intuition about how the hard correction works during pre-training, we collect the statistics of the hard corrections in two perspectives: 1) prediction-level: the proportion of corrected predictions when they mismatch the gold labels; 2) iteration-level: the proportion of iterations when the correction happens. As the training corpus is relatively large, we use the wikitext-2-raw-v1 corpus (Merity et al., 2016) for efficient validation as suggested by Transformers [6]. We use the pre-trained checkpoints with hard correction on the backbones of BERT-base and BERT-large models for the analysis.

Table 8 shows the statistics, from which we have the following observations:

1) The correction ratio is around 1.0%-2.0% in token-level and around 6.0%-7.0% in iteration-level.

2) As the training goes on, the correction ratio increases, indicating that our method would gradually play a more important role when the training goes on, which supports our hypothesis.

3) The correction ratio in larger models would be higher than the base models, which indicates larger models would be more likely to encounter false negatives.

### A.4 HOW CAN WE MEASURE THE SEVERITY OF THE FALSE NEGATIVE PROBLEM?

As indicated in the Section A.3, we see that our method is gradually playing a more important role when the training goes on. Since the training examples are based on random masking, the PrLMs are thus forced to be trained on low-quality samples.

As the saying goes, "The rotten apple injures its neighbors", training on random low-quality examples would bring training bias from meaningless data, so it needs to be corrected with more data and results in more cost of resource and time. Our methods can be regarded as the training correction to help the model train on more "true samples"; thus, they would improve the training efficiency and help the model to get rid of adversarial attacks like diversity distraction and synonym substitution.

---

[6]https://github.com/huggingface/transformers/tree/master/examples/pytorch/language-modeling

