# OpenReview forum: "Language Model Pre-training on True Negatives"
_ICLR.cc/2022/Conference — ICLR 2022 Submitted_

### Official Review · Reviewer_1R2h · 2021-10-31

**Correctness:** 3
**Technical Novelty And Significance:** 2
**Empirical Novelty And Significance:** 2
**Recommendation:** 5
**Confidence:** 4

**Main Review:**

Pros:

- As summaried above, this paper presents technically sound methods (i.e., SR and HC) to mitigate false negative problem in masked language models.
- Besides the main evaluation to beat baseline performance on natural language understanding tasks, this paper also includes more insightful analyses into the proposed methods, from both quantitative and qualitative perspectives.


Cons and Questions:

- The motivation of this paper is not that convincing to me. I fully agree "it is critical to ensure the auto-constructed data is true enough" but how can we measure the severity of "false negative problem"? Given the 2nd & 3rd examples in Table 1, although both "Ground-truth" and "Prediction" can fit the masked tokens, there are still subtle differences between them especially in long contexts (up to 512/1024 tokens during pretraining vs. a short sentence in the examples).
- There are also some weaknesses in the proposed pre-training objectives. As for SR, the regularization is only applied to embedding layer. As for HC, Wordnet can only provide a very coarse Symset, which may introduce error propagation. Can author give more empirical studies to demonstrate the successes of these objectives?
- Why did the author only apply the proposed pre-training objectives on the top of ELECTRA instead of standard MLM, e.g., RoBERT. The latter should be a better testbed for the objectives targeting problems in MLM.
- Can authors discuss the relation of this work to a recent distillation paradigm, i.e., distill a small LM to large LM for accelerated convergence and promoted performance, which is mainly attributed to label smoothing.


**Summary Of The Paper:**

This paper focuses on the issue of misconceived false negative predictions and encourage pre-training language models on true negatives or more true negatives to ensure the auto-constructed data in masked language modeling (MLM) is true enough. A motivation behind this is that  false negatives may potentially hurt the pre-training in both efficiency and robustness to a great extent and this vulnerability of pre-trained language has never been studied before. In this paper, the authors propose an enhanced pre-training approach to counteract misconceived negatives.

Based on ElECTRA, the authors present two alternative pre-training objective auxilary to MLM one, which aims to make the model focus on true negatives during MLM. The first one, dubbed as soft regularization (SR), measures the distribution similarity between the predicted token and the original one, to smooth the tough cross-entropy by minimizing the semantic distances. The other, named as hard correction (HC), hinder the gradient propagation of the false negative samples to further avoid training with false negative predictions.

**Summary Of The Review:**

Technically sound methods but unconvincing motivation and ill-considered methods; More empirical studies needed.

---

> ### Author Response · Authors · 2021-11-12
> **Response to Reviewer 1R2h**
>
> Thanks for your time and valuable feedback.
>
> 1.  **How can we measure the severity of \"false negative problem\"?**
>
> As indicated in the analysis in General Response #4, we see that our method is gradually playing a more important role when the training goes on. Since the training examples are based on random masking, the PrLMs are thus forced to be trained on low-quality samples.
>
> As the saying goes, "The rotten apple injures its neighbors.", training on random low-quality examples would bring training bias from meaningless data, so it needs to be corrected with more data and results in more cost of resource and time. Our methods can be regarded as the training correction to help the model train on more "true samples"; thus, they would improve the training efficiency and help the model to get rid of adversarial attacks like diversity distraction and synonym substitution.
>
> 2.  **Weaknesses in the proposed pre-training objectives.**
>
> For SR, the regularization is only applied to the embedding layer because the embedding layer is the most fundamental and stable layer. Optimizing the embedding layer would possibly lead to a more severe influence of the model training and help the model learn semantics between words better as indicated by \[1\].
>
> > \[1\] SMART: Robust and Efficient Fine-Tuning for Pre-trained Natural Language Models through Principled Regularized Optimization. ACL 2020.
>
> For HC, the hard correction would be resistible to error propagation because it is a post-processing technique. As described in the last paragraph of Section 3.2, if the predicted token is in the shortlist, the correction will be activated by simply ignoring this prediction before feeding to cross-entropy loss function: the corresponding gold label id will be replaced by -100 (default ignore_index in nn.CrossEntropy using PyTorch), which means this token is not required to predict anymore. The only effect is the ignorance of a predicted token, which hardly does any harm.
>
> 3.  **Why apply the proposed objectives ELECTRA instead of standard MLM, e.g., RoBERT.**
>
> We select ELECTRA because it can enjoy a good training efficiency. We have already experimented on BERT-base and BERT-large, where our methods have achieved consistent gains. Please see the General Response #3. We did not use RoBERTa because RoBERTa uses a ByteLevelBPE tokenizer that would be incompatible with the hard correction which requires a word-wise vocabulary.
>
> 4.  **Can authors discuss the relation of this work to a recent distillation paradigm?**
>
> Yes, we will add the discussion in the next version.

---

### Official Review · Reviewer_QW2m · 2021-11-03

**Correctness:** 2
**Technical Novelty And Significance:** 2
**Empirical Novelty And Significance:** 2
**Recommendation:** 3
**Confidence:** 4

**Main Review:**

The motivation of reducing false negative examples is reasonable but still does not convince me that this is necessary in pretraining, given that the main goal of pretraining is to learn good contextualized representation that could be useful for various downstream tasks. The authors claim the false negative examples are harmful to pretraining, but it is not fully supported by the experiments in the paper. The intuition of soft regularization and hard correction is somehow ad-hoc and may also not hold under some circumstances. For example, when the model is not well trained, minimizing the distance between the predicted token and the ground-truth token could be misleading if the prediction is incorrect. The synonyms from the dictionary could also be wrong without considering the given context. In equation (6), it is unclear whether the embedding e_k is token embedding or contextualized representation, and from generator or discriminator.

Some experimental results are problematic and not convincing.
1)	The proposed methods are only applied to continual training of a well-trained ELECTRA model. This may indicate they are not suitable for early-stage pretraining, as the assumptions may not hold. This is a great limitation as a pretraining strategy, as it is not clear at which training steps one can use it. Moreover, as a fair baseline, ELECTRA should also be continually trained with the same 200k steps as the proposed models.
2)	The results in Table 2 are inconsistent with Table 1 (ELECTRA_small and ELECTRA_base on GLUE test set). Some important baseline such as RoBERTa, XLNet, DeBERTa are not included either. Moreover, since all the experiments are conducted on ELECTRA small and base model, it is unknown whether the proposed methods work on larger scale models and other types of pretrained models such as BERT.
3)	In Table 6, what is the baseline system, BERT or ELECTRA_small?


**Summary Of The Paper:**

This paper proposes to avoid false negative examples in language model pretraining via modifying the training objective. Two methods, referred to as soft regularization and hard correction, are proposed to reduce the false negatives based on continual training on the ELECTRA model. Experiments on the GLUE benchmark show that the continual trained model with the two methods can both improve the performance on downstream NLU tasks. Experiments on SQuAD show the methods also help with model robustness.
This work provides ways to adjust the pre-training objectives when the model predictions are correct but do not match the ground-truth tokens. This could benefit the performance on downstream task in some cases. But the limited experiments in the paper do not assure that this is a general approach in pretraining that can be widely adopted.

**Summary Of The Review:**

The methods proposed in this paper have clear theoretical and empirical limitations. The experiments do not fully support the claims by the authors. My current recommendation to this paper is reject.

---

> ### Author Response · Authors · 2021-11-12
> **Response to Reviewer QW2m**
>
> Thanks for your time and valuable feedback.
>
> 1.  **Only applied to continual training of a well-trained ELECTRA model.**
>
> Not true. As described in Section 4.1, the small models are pre-trained (from scratch) for 1000\$k\$ steps. Only the base models are continually pre-trained due to a lack of computation resources.
>
> As clarified "To save computation, like previous studies (Dong et al., 2019), we continue training base models for 200\$k\$ steps using the pre-trained weights as initialization."
>
> The baselines are trained to the same steps for fair comparison following the standard procedure in Dong et al., (2019).
>
> 2.  **The results in Table 2 are inconsistent with Table 1.**
>
> It is reasonable. For the small model, as clarified in the official [GitHub repository of ELECRA](https://github.com/google-research/electra): "Unfortunately, the data we used in the paper is not publicly available, so we will use the OpenWebTextCorpus", the reproduced results (in Table 1) would have slight difference with the reported ones in the ELECTRA paper (average score of 78.3 vs. 78.0). For the large model, since they are based on continued training, we report the ELECTRA-base baseline with extra 200\$k\$ steps for a fair comparison, thus the numbers would be higher than the public ones.
>
> We did not include the RoBERTa, XLNet, DeBERTa results because their papers did not report the GLUE test results of single models for fair comparison, but use ensemble models and heuristic fine-tuning strategies instead. Therefore, we compare the models with compariable size, under the standard test setting.
>
> 3.  **In Table 6, what is the baseline system, BERT or ELECTRA_small?**
>
> It is ELECTRA_small.
>
> **Other comments:**
>
> 1)  "when the model is not well trained, minimizing the distance between the predicted token and the ground-truth token could be misleading if the prediction is incorrect."
>
> In the very early stage, the original loss dominates the training process, which would be several times larger than the regularization (which basically falls into 0-1). Therefore, the regularization would affect the performance when the iterations have relatively high accuracy.
>
> In addition, the idea of soft regularization is based on the hypothesis that the predicted tokens should have a semantic relationship with the gold ones to some extent, which is closely related to the idea of TransE \[1\]. Various studies \[2\]\[3\]\[4\] have also supported the hypothesis.
>
> > \[1\] Translating Embeddings for Modeling Multi-relational Data. NeurIPS 2013.
> >
> > \[2\] Structural Pre-training for Dialogue Comprehension. ACL 2021.
> >
> > \[3\] Dialogue Summarization with Supporting Utterance Flow Modeling and Fact Regularization. Knowledge-Based Systems.
> >
> > \[4\] Data-dependent Gaussian Prior Objective for Language Generation. ICLR 2020.
>
> 2)  "it is unclear whether the embedding e_k is token embedding or contextualized representation, and from generator or discriminator."
>
> It is token embedding (as we described in Section 3.2: "we fetch their token representations from the model\'s embedding module") from the discriminator (please note that the embeddings of the generator and the discriminator are tied following the official implementation).

---

### Official Review · Reviewer_dhbH · 2021-11-08

**Correctness:** 2
**Technical Novelty And Significance:** 2
**Empirical Novelty And Significance:** 2
**Recommendation:** 3
**Confidence:** 4

**Main Review:**

Strengths
- S1 - The idea of handling true negatives for ELECTRA pre-training is an interesting idea.
- S2 - Experimental results show that proposed pre-training objectives can improve the performance of ELECTRA.

Weaknesses and Questions:
- W1 - Quiet unsure about the experimental settings. Seems SR and HC are continually pre-training objectives. Then, are baselines (ELECTRA_small and ELECTRA_base) also continually pre-trained with original ELECTRA pre-training objectives for the same steps of SR and HC? If not, the results are not convincing.
- W2 - Unclear the performance improvement is solely from handling true negatives. what about (1) random (2) false negatives?
- Q1 - It would be great if the paper could provide the correction statistics of HC.
- Q2 - ELECTRA-style token-level discriminative pre-trained models are showing good performance in token-level classification tasks such as NER. It would be great if we could see that performance as well.

**Summary Of The Paper:**

The premise of the paper is that true negative examples should be considered in ELECTRA-style token-level discriminative pre-training. The framework uses cosine similarity to find true negatives and softly regularize with semantic distance, or finding true negatives by synonyms and simply ignoring the prediction before feeding to the loss function. Results on GLUE and SQUAD with ELECTRA-based models show that handling true negatives during pre-training improves performance.

**Summary Of The Review:**

Although the paper shows good results compared to the baseline, current writing is not convincing.
It needs more details and extensive analysis to support the claim.

---

> ### Author Response · Authors · 2021-11-12
> **Response to Reviewer dhbH**
>
> Thanks for your time and valuable feedback.
>
> 1.  **W1 - Seems SR and HC are continually pre-training objectives.**
>
> Not exactly. SR and HC are verified effective for both training scratch and continued training. As described in Section 4.1, the small models are pre-trained (from scratch) for 1000\$k\$ steps. Only the base models are continually pre-trained due to a lack of computation resources.
>
> As clarified "To save computation, like previous studies (Dong et al., 2019), we continue training base models for 200\$k\$ steps using the pre-trained weights as initialization.", yes, the baselines are trained to the same steps for fair comparison following the standard procedure in Dong et al., (2019).
>
> 2.  **W2 - Unclear the performance improvement is solely from handling true negatives.**
>
> It seems like a misunderstanding. Our work is to encourage pre-training language models on true negatives, by correcting the harmful gradient updates subject to false negative predictions. It is not handling true negatives.
>
> 3.  **Q1 - It would be great if the paper could provide the correction statistics of HC.**
>
> Yes, we have collected the statistics of the hard corrections. Please see the General Response #4.
>
> 4.  **Q2 - Performance on the token-level classification tasks such as NER.**
>
> We are experimenting with the NER task and will add the results later. Please note pre-trained models are supposed to work effectively on most tasks in general, instead of designed for specific tasks like NER -- It might not be a good testbed for evaluating PrLMs without further enhancements. Actually, to verify the generality, we have already verified the method on 9 tasks from GLUE and SQuAD, along with robustness evaluations against adversarial attacks.

---

### Official Review · Reviewer_jW6M · 2021-11-09

**Correctness:** 3
**Technical Novelty And Significance:** 2
**Empirical Novelty And Significance:** 3
**Recommendation:** 5
**Confidence:** 3

**Main Review:**

*Strengths:
1. The false negative problem tackled in this paper is critical for pretraining and seems to be ignored by the mainstream. I believe improvement on this problem can lead to better representation models, and can be inspiring to other pretraining objectives.
2. The method brings consistent gains compared to fair baselines on two standard benchmarks.

*Weaknesses:
1. My main concern is about the proposed methods in this paper. First, for the Soft Regularization method, it is not clear how the loss L_reg backpropagates through the model. Is the author only training the two embeddings? It doesn't make sense to me that this can improve the model.
2. For the Hard Correction method, it looks to me like the author is just using a synonym set as the target labels. Despite of the good empirical results, the solution itself lacks novelty. Also, although the WordNet synonyms can mitigate false negative problem, it can also brings new false positives.
3. The author mainly compares their method with the original pretrained model itself, rather than other methods in solving false positives. I am curious if add label smoothing during training can also bring gains and how it compares with the proposed method in this paper.

**Summary Of The Paper:**

This paper identifies the largely neglected false negative problem in training or pretraining language models. For many cases in training LM, the training objective only uses one single token as the target, while treating other meaningful tokens as equally negative. To tackle this problem, the author proposes two methods: 1) the Soft Regularization method minimizes the embedding distance between the predicted and gold tokens; 2) the Hard Correction employs WordNet and maximizes all the synonyms of the gold token. They demonstrate consistent gains and better robustness by applying this to pretraining ELECTRA, and testing on GLUE and SQuAD downstream tasks.

**Summary Of The Review:**

This paper studies a critical problem of false negatives in pretraining language models and demonstrates good results. But the proposed solution lacks enough novelty and might have some technical flaws.

---

> ### Author Response · Authors · 2021-11-12
> **Response to Reviewer jW6M**
>
> Thanks for your time and valuable feedback.
>
> 1. **About the soft regularization method**
>
> This loss can backpropagate as expected. Please see the General Response #1.
>
> It proceeds by a similar way of training the discriminator using the predicted tokens (still "pred_toks") from the generator, where the fundamental embeddings will be optimized.
>
> Yes, it estimates the similarity of the embeddings. It is based on the hypothesis that the predicted tokens should have a semantic relationship with the gold ones to some extent, which is closely related to the idea of TransE \[1\]. The similar calculation has been applied to various studies such as \[2\]\[3\]\[4\], which verified the effectiveness.
>
> > \[1\] Translating Embeddings for Modeling Multi-relational Data. NeurIPS 2013.
> >
> > \[2\] Structural Pre-training for Dialogue Comprehension. ACL 2021.
> >
> > \[3\] Dialogue Summarization with Supporting Utterance Flow Modeling and Fact Regularization. Knowledge-Based Systems.
> >
> > \[4\] Data-dependent Gaussian Prior Objective for Language Generation. ICLR 2020.
>
> 2.  **WordNet synonyms can also bring new false positives.**
>
> The hard correction would not bring false positives because it is a post-processing technique. Please see the General Response #2.
>
> 3.  **Other methods in solving false positives.**
>
> Yes, as you figured out, "The false negative problem tackled in this paper is critical for pretraining and seems to be ignored by the mainstream", there is no other work on pre-training that has investigated the issue before. We agree that it would be beneficial to compare with label smoothing. We will add the comparison once the experiments are finished. Thanks for your constructive comment.

---

### Official Review · Reviewer_syob · 2021-11-09

**Correctness:** 3
**Technical Novelty And Significance:** 2
**Empirical Novelty And Significance:** 2
**Recommendation:** 5
**Confidence:** 4

**Main Review:**

Strengths:
1) The paper identified the false negatives issue in PrLMs and could possibly inspire other researcher to work on this.
2) The experiments are conducted on two benchmarks and the proposed methods show consistent gains over a strong baseline model.

Weakness:
1) For the soft regularization in equation (6), I am not sure how this loss can be backpropagated since the token prediction step is a discrete argmax function.
2) For the hard correction, it is unclear whether the WordNet will also bring false positive, i.e. a synonym which should not fill in the mask.
3) The experiments only cover one pretrained language model - ELECTRA. Could this method be applied on other MLM PrLMs such as RoBERTa?
4) I also expect some statistics of the hard corrections in the pretraining experiment.

Presentation Issues:
1) In table 2, the test set result of ELECTRA_small on QQP is the best but the number is not bold.

**Summary Of The Paper:**

The paper identified the false negative issue in the pre-trained language models and proposed two methods to counteract it. The first method replaces the cross-entropy loss in the standard MLM objective with the cosine distance between the representations of predicted token and the ground truth. The second method prunes the gradients from the false positive tokens, where such tokens are identified from synonym alternatives of WordNet. Empirical results on GLUE and SQuAD show that the proposed solution improves ELECTRA-small and ELECTRA-base models for both performance and robustness.

**Summary Of The Review:**

The paper proposed two simple methods for a novel issue in PrLMs. The empirical results are promising in two benchmarks with ELECTRA model. However, more technical description, empirical analysis, and experimental result are still needed to fully support the claim of this paper.

---

> ### Author Response · Authors · 2021-11-12
> **Response to Reviewer syob**
>
> Thanks for your time and valuable feedback.
>
> 1.  **Backpropagation of the soft regularization.**
>
> This loss can backpropagate as expected. Please see the General Response #1.
>
> 2.  **Whether the WordNet will also bring false positive.**
>
> The hard correction would not bring false positives because it is a post-processing technique. Please see the General Response #2.
>
> 3.  **Could this method be applied on other MLM PrLMs?**
>
> We have already experimented on BERT-base and BERT-large, where our methods have achieved consistent gains. Please see the General Response #3. We did not use RoBERTa because RoBERTa uses a ByteLevelBPE tokenizer that would be incompatible with the hard correction which requires a word-wise vocabulary.
>
> 4.  **Statistics of the hard corrections.**
>
> Yes, we have collected the statistics of the hard corrections. Please see the General Response #4.
>
> 5.  **Presentation Issues.**
>
> Thanks for the correction. We have fixed it and will update the paper later.

---

### Author Response · Authors · 2021-11-12
**General Response**


Dear reviewers,

Thank you so much for your time and the constructive reviews. We appreciate your recognition of the core idea of dealing the false negative problem, consistent gains compared to fair baselines, and insightful analyses. We are working hard on various experiments as suggested and will update the paper accordingly.

First, we address three general concerns. All these concerns are raised just due to the limited pages allowed by ICLR, and we have to skip the very details. We are glad none of these concerns root from our proposed method itself on either implementation or motivation. Thus they are easily clarified as below.

1.  **Backpropagation of the soft regularization**

The backpropagation is not an issue. It proceeds by a similar way of training the discriminator using the predicted tokens from the generator, where the fundamental embeddings of the discrimator will be optimized. Such a practice has been successfully applied in ELECTRA [1].

> [1] ELECTRA: Pre-training Text Encoders as Discriminators Rather Than Generator. ICLR 2020.

The loss function corresponds to the similarity of the token embeddings. We paste the main implementation below for reference.  Given the prediction token ids (pred_toks) and the gold token ids (gold_ids), the loss is calculated by:

```
pred_embeddings = self.discriminator.electra.embeddings.word_embeddings(pred_toks)
gold_embeddings = self.discriminator.electra.embeddings.word_embeddings(gold_ids)
syn_sim = F.cosine_similarity(pred_embeddings, gold_embeddings)
syn_inds = torch.nonzero(syn_sim)
syn_sim = syn_sim[syn_inds]
syn_loss = 1. - syn_sim
syn_loss = torch.mean(syn_loss)
```
Our full codes and models will be publicly available.

2.  **Would the hard correction bring false positives?**

Not a chance. The hard correction would not bring false positives because it is a post-processing technique. As described in the last paragraph of Section 3.2, if the predicted token is in the shortlist, the correction will be activated by simply ignoring this prediction before feeding to cross-entropy loss function: the corresponding gold label id will be replaced by -100 (default ignore_index in nn.CrossEntropy using PyTorch), which means this token is not required to predict anymore.

3.  **Could this method be applied on other MLM PrLMs?**

Sure. We have already experimented on BERT-base and BERT-large. To save resources, we continue training the baselines and our methods for 200\$k\$ steps. We did not use RoBERTa because RoBERTa uses a ByteLevelBPE tokenizer that would be incompatible with the hard correction which requires a word-wise vocabulary.

| Model                         | CoLA  | SST   | MRPC  | STS   | QQP   | MNLI  | QNLI  | RTE   | Avg   |
| ----------------------------- | ----- | ----- | ----- | ----- | ----- | ----- | ----- | ----- | ----- |
| BERT$_{\rm{base}}$            | 61.09 | 93.00 | 86.76 | 87.09 | 90.79 | 84.72 | 91.42 | 67.87 | 82.84 |
| BERT$^{\rm{SR}}_{\rm{base}}$  | 61.17 | 93.46 | 88.97 | 87.45 | 90.93 | 84.83 | 91.62 | 68.59 | 83.38 |
| BERT$^{\rm{HC}}_{\rm{base}}$  | 62.88 | 93.23 | 87.50 | 87.41 | 90.92 | 84.92 | 91.54 | 69.31 | 83.46 |
| BERT$_{\rm{Large}}$           | 61.67 | 93.69 | 88.48 | 90.14 | 91.30 | 86.74 | 92.37 | 72.92 | 84.67 |
| BERT$^{\rm{SR}}_{\rm{large}}$ | 62.26 | 94.15 | 89.22 | 90.12 | 91.41 | 87.01 | 92.82 | 74.01 | 85.13 |
| BERT$^{\rm{HC}}_{\rm{large}}$ | 62.34 | 93.35 | 88.97 | 90.48 | 91.46 | 86.96 | 92.95 | 73.65 | 85.02 |

The table above shows the preliminary results on the GLUE dev sets. We see that our methods still achieve consistent gains.

---

> ### Author Response · Authors · 2021-11-12
> **General Response (cont.)**
>
>
> 4.  **Statistics of the hard corrections**
>
> We have collected the statistics of the hard corrections in two perspectives: 1) *prediction-level:* the proportion of corrected predictions when they mismatch the gold labels; 2) *iteration-level:* the proportion of iterations when the correction happens.
>
> As the training corpus is relatively large, we use the *[wikitext-2-raw-v1](https://huggingface.co/datasets/wikitext)* corpus for quick validation as suggested by [Transformers](https://github.com/huggingface/transformers/tree/master/examples/pytorch/language-modeling). We use the pre-trained checkpoints with hard correction on the backbones of BERT-base and BERT-large models for the analysis.
>
> | Checkpoint (base) | Iteration-level | Prediction-level | Checkpoint (large) | Iteration-level | Prediction-level |
> | ----------------- | --------------- | ---------------- | ------------------ | --------------- | ---------------- |
> | 6.25\%            | 6.90\%          | 1.31\%           | 6.25\%             | 7.46\%          | 1.50\%           |
> | 12.5\%            | 6.96\%          | 1.34\%           | 12.5\%             | 7.58\%          | 1.55\%           |
> | 25.0\%            | 6.97\%          | 1.36\%           | 25.0\%             | 7.31\%          | 1.49\%           |
> | 50.0\%            | 7.05\%          | 1.36\%           | 50.0\%             | 7.46\%          | 1.56\%           |
> | 80.0\%            | 7.06\%          | 1.40\%           | 80.0\%             | 7.38\%          | 1.57\%           |
> | 100.0\%           | 7.07\%          | 1.41\%           | 100.0\%            | 7.44\%          | 1.60\%           |
>
> **Checkpoint means the checkpoint saved at the specific training steps (%).*
>
> The table above shows the results, from which we have the following observations:
>
> 1)  The correction ratio is around 6.0%-7.0% in iteration-level and around 1.0%-2.0% in token-level.
>
> 2)  As the training goes on, the correction ratio increases, indicating that our method would gradually play a more important role when the training goes on, which supports our hypothesis.
>
> 3)  The correction ratio in larger models would be higher than the base models, which indicates larger models would be more likely to encounter false negatives.
>
> For the other concerns and questions, please see the point-to-point response below.

---

### Author Response · Authors · 2021-11-12
**Submission Update**

We thank all the reviewers so much for the valuable comments on improving the quality of this work. We have updated the paper according to the feedback and our latest evaluations. The main modifications are highlighted with blue color in the newly updated PDF.

The revision primarily includes:

1. We add a discussion about the recent distillation paradigm. (Section 2).

2. We add more details about the process of soft regularization (Section 3.2).

3. We clarify the pre-training setup (Section 4.1).

4. We clarify the reliability of the hard correction (Appendix A.1).

5. We add the application to other PrLMs (Appendix A.2).

6. We add the statistics of the hard corrections (Appendix A.3).

7. We add a discussion of the severity of the "false negative" problem  (Appendix A.4).

Thanks again for your feedback. We hope the response and revision can address the concerns. If you have any further comments or suggestions, we will be happy to respond.

---

### Decision · Program_Chairs · 2022-01-20

**Decision:**

Reject

**Comment:**

This paper gets decent performance gains (~2% on GLUE) by soft regularization to make negatives closer to positives in contrastive learning and hard correction of too-close negatives to at least avoid synonyms. These are useful ideas which to some extent build on the simple technique of ELECTRA (controlling the size of the generator MLM in Electra encourages the negatives to in general be "close but not too close", right?). As such, the paper is correct and provides potentially useful gains, but it appears a quite small adjustment of existing techniques, and in addition the use of WordNet is fairly brittle (and its similarity calculations do not consider context at all).

The authors should be commended for the thorough job they did at updating their paper to address particular questions and concerns of reviewers, and useful new information emerged. Relative to the question of whether this method can be applied with other MLMs, the new Appendix A results do show that the answer is Yes, but the gains turn out to be much more modest (~0.5% on GLUE). However, ultimately, while this is all useful information and side experiments, these improvements just can't fix the key problem that all the reviewers felt that this paper does not provide sufficient "Technical Novelty and Significance". As such without bigger new ideas, this improved paper would probably be best as a good workshop paper.

My recommendation is that this paper not be accepted to ICLR 2022 on the basis of its limited technical novelty and significance.